# Vascular Calcification: In Vitro Models under the Magnifying Glass

**DOI:** 10.3390/biomedicines10102491

**Published:** 2022-10-06

**Authors:** Elisa Ceccherini, Antonella Cecchettini, Ilaria Gisone, Elisa Persiani, Maria Aurora Morales, Federico Vozzi

**Affiliations:** 1Institute of Clinical Physiology, National Research Council (CNR), 56124 Pisa, Italy; 2Department of Clinical and Experimental Medicine, University of Pisa, 56126 Pisa, Italy

**Keywords:** vascular calcification, VSMCs, in vitro models, anticalcifying agents

## Abstract

Vascular calcification is a systemic disease contributing to cardiovascular morbidity and mortality. The pathophysiology of vascular calcification involves calcium salt deposition by vascular smooth muscle cells that exhibit an osteoblast-like phenotype. Multiple conditions drive the phenotypic switch and calcium deposition in the vascular wall; however, the exact molecular mechanisms and the connection between vascular smooth muscle cells and other cell types are not fully elucidated. In this hazy landscape, effective treatment options are lacking. Due to the pathophysiological complexity, several research models are available to evaluate different aspects of the calcification process. This review gives an overview of the in vitro cell models used so far to study the molecular processes underlying vascular calcification. In addition, relevant natural and synthetic compounds that exerted anticalcifying properties in in vitro systems are discussed.

## 1. Introduction

Vascular calcification (VC) is a major complication of different pathologies such as atherosclerosis, diabetes, hypertension, and chronic kidney disease (CKD). It is characterized by massive calcium salt deposition in the vascular wall and decreases blood vessel elasticity [1]. The stiffness of vessel walls represents a key factor in pathogenesis and plays a role in overall morbidity and mortality. The increase in calcium and phosphate levels (Pi) also pushes vascular smooth muscle cells (VSMCs) towards differentiation into osteoblast-like cells [2]. VSMCs are characterized by switching from a differentiated “contractile” into a dedifferentiated “synthetic” proliferative phenotype (Figure 1). This transformation is accompanied by decreased expression of contractile markers (e.g., smooth muscle α-actin (α-SMA), smooth muscle 22α (SM22α), smooth muscle myosin heavy chain 11 (SM-MHC), calponin (CCN1), smoothelin (SMTN), and myosin light chain (MYL)) and increased synthetic markers expression (S100A4, KLF4, vimentin, osteopontin (OPN)) [3]. In addition, dedifferentiated VSMCs express higher osteoblast-like markers, such as runt-related transcription factor 2 (Runx2), zinc finger transcription factor (Osterix), and muscle segment homeobox 2 (MSX2) [4]. Mechanisms such as inflammation, oxidative stress, loss of mineralization inhibitors, and increased extracellular matrix (ECM) remodeling contribute to the phenotypic transformation of VSMCs [5]. These processes prime other changes in synthetic VSMCs, leading to osteogenic differentiation and an osteoblastic-like or calcified phenotype. Aside from differentiated VSMCs, different cell types are associated with VC, including mesenchymal, endothelial, hematopoietic osteoprogenitor cells, and myeloid cells with osteogenic and calcifying potential [6,7].

The molecular mechanisms underlying VC are complex and not yet completely elucidated [8]. Despite significant progress in pharmacological treatments, clinical complications of VC remain one of the leading causes of death. Considering the clinical and prognostic value of VC and the often inadequate transferability of animal data to the clinical human setting, human in vitro models may represent a valuable platform for performing investigations on the pathogenesis of VC. These systems grant co-culture of different cell types, such as VSMCs, endothelial cells (ECs), and bone marrow-mesenchymal cells (BM-MSCs), overcoming the limitations of monolayer cell cultures and allowing simulation of the complexity of the arterial tissue closely.

## 2. Relevant Mechanisms in Vascular Calcification

VC is a complex biological process regulated by multiple factors, including phenotypic conversion of VSMCs, calcium and Pi metabolic homeostasis, inflammation, oxidative stress, autophagy, and vesicle release [9,10,11,12].

In vitro studies have shown that increased Ca^2+^ concentration can reduce the calcium-sensing receptor expression and increase mineral deposition in VSMCs [13]. Intracellular calcium overload leads to a series of disorders in superoxide metabolism, resulting in oxidative stress [14], a key mechanism of the osteogenic phenotype conversion of VSMCs [15,16]. In addition, chronic inflammation can promote and further exacerbate VC progression through multiple cytokines, including IL-6, IL-1β, and TNF-α, via JAK/STAT3, AMPK, cAMP, and NF-kB pathways [17,18].

It has also been demonstrated that the expression of growth arrest-specific gene 6 (Gas6), a member of the vitamin K-dependent protein family, and its receptor Axl is involved in cell survival in a range of cell types and is down regulated during VC [19]. To maintain cell survival, the binding of Gas6 to Axl induces phosphatidylinositol 3-OH kinase (PI3K) activation and subsequent Akt activation. It has recently identified the Akt signaling as a critical regulator during osteogenic differentiation of VSMCs [20]. In this study, extracellular vesicles derived from uremic toxin-treated endothelial cells in vitro or plasma-derived extracellular vesicles from uremic rats induced the osteogenic transdifferentiation of VSMCs. These pro-calcifying effects were neutralized by pharmacological inhibition of AKT, confirming its pivotal role in the vascular calcification process.

Autophagy is a cellular survival mechanism necessary for maintaining organelle quality control, acting in parallel with the ubiquitin-proteasome degradation pathway to suppress the accumulation of polyubiquitinated and aggregated proteins. Autophagy plays a protective role in the progression of certain human disorders, including VC [21]. Dai and colleagues showed that Pi increased ROS levels and induced autophagy in cultured VSMCs [22]. The autophagy inhibitor 3-methyladenine and siRNA knockdown of the autophagy protein 5 (Atg5) significantly aggravated Pi-induced calcium deposition in vitro. In the setting of VC, autophagy represents a protective mechanism for counteracting the calcification process.

Recently, the inhibition of the mammalian target of rapamycin (mTOR) was demonstrated as a prominent pathway that modulates several aspects of mesenchymal stem cells (MSC) physiology, including induction of a VSMC differentiated state [23]. The positive effect of mTOR inhibition on VSMC differentiation could be related to Akt activation and contractile protein expression in an insulin-like growth factor I-dependent manner by relieving S6K1-dependent negative regulation of insulin receptor substrate-1 [24]. In addition, mTOR signaling regulates VSMC migration [25]. The cyclin-dependent kinase inhibitor, p27Kip1, regulates VSMC proliferation and migration. In a quiescent state, VSMCs show elevated levels of p27Kip1 that block proliferation and migration and inhibit neointimal hyperplasia [26]. Upon injury, the p27Kip1 protein is downregulated by activating the mTOR signaling, promoting VSMC proliferation and migration.

In the last decade, researchers turned interest to sirtuin 1 (sirt1), which is highly expressed in the vasculature and protects against atherosclerosis [27] and VC [28]. Sirt1 is a nicotinamide adenine dinucleotide (NAD+)-dependent deacetylase that regulates vascular cell aging, energy metabolism, apoptosis, genomic stability, and stress responses [29]. Cultured aortas of mice with sirt1 knockdown showed accelerated medial calcification induced by inorganic phosphate [30]. Moreover, sirt1 downregulation promoted VSMC senescence and calcification under osteogenic conditions by inhibiting p21 and the osteogenic transcription factor Runx2 [28].

## 3. How to Investigate Vascular Calcification? From Old to Modern In Vitro Models

### 3.1. Monolayer Cell Cultures

The simplest in vitro model is based on the monoculture of VSMCs. Although some authors demonstrated spontaneous calcification within six days of culture in VSMCs isolated from hypertensive rats [31], most cells require external stimuli to calcify. The experimental conditions commonly used to induce calcification are presented in Table 1 and are generally maintained for at least seven days to obtain intracellular calcium deposition.

A recent study provides interesting information regarding the role of Pi and calcium additives on calcification induction in aortic SMCs (AoSMCs) [40]. Calcification was not observed using low Pi and high concentrations of calcium ions, confirming the essential role of high Pi levels in this process. In addition, the ability to induce intracellular calcium accumulation has been tested in different Pi donors (acidic, neutral, and basic). The neutral phosphate donors have proven the best for generating calcium accumulation in high calcium conditions (i.e., 2.4 mM). Holmar and colleagues have also demonstrated that incubating VSMCs with high levels of Pi and calcium ions increases calcification and promotes trans-differentiation into osteoblast-like cells [41]. In these experiments, an evident overexpression of osteogenic markers such as osteocalcin and transcription factor core-binding factor-1 (Cbfa-1), essential for osteoblast differentiation, has been observed. The effect of hyperphosphatemia is ruled by a sodium-dependent phosphate co-transporter (NPC) that mediates the entry of Pi into vascular cells leading to an imbalance in calcium-Pi homeostasis. Adding NPC-specific inhibitors (phosphonoformic acid or arsenate) to the culture medium inhibited Pi uptake in VSMCs and the expression of osteocalcin and Cbfa-1 markers. In this regard, the monoculture of AoSMCs exposed to physiologic levels of Pi revealed normal cell growth, and they respond to higher Pi concentrations (up to 2 mmol/L) by increasing pro-mineralization factors and deposition of calcium (positive von Kossa staining). In addition, transmission electron microscopy and electron diffraction verified the presence of apatite crystals, matrix vesicles, and calcified collagen fibers. The experimental setting of monolayer cell culture is easily arranged; however, it cannot assess the interaction between different cell types and their signaling pathways.

### 3.2. ECs and VSMCs Co-Cultures

ECs and VSMCs are the main cellular components of vascular parenchyma and play essential roles in vascular homeostasis [42]. The interactions between vascular endothelium and smooth muscle cells are required for a healthy vessel. Co-culture systems have been used to investigate the interconnections between these two different cell types and are based on transwell assay [43]: this system allows ECs and VSMCs to communicate via soluble mediators (e.g., growth factors and cytokines), mimicking an environment similar to the physiological one. They are suitable for investigating mechanisms underlying VC and atherosclerosis.

The AoSMCs, isolated from spontaneously hypertensive rats with endothelial dysfunction and co-cultured with ECs, exhibited exacerbated calcification and higher expression of matrix metalloproteinase-2 (MMP-2) and matrix metalloproteinase-9 (MMP-9) than those cultured without ECs, confirming the ability of ECs to promote calcification in this setting [44]. Evensen and colleagues showed that VSMCs secrete growth factors that stimulate the ECs proliferation, differentiation, migration, and the expression of proteins involved in the deposition of extracellular matrix proteins [45]. VSMCs promote ECs adhesion by modulating the polymerization of the microtubule cytoskeleton [46]. Xiang and colleagues isolated ECs and VSMCs from WT mice and apoE-/-mice and used these cells as an in vitro integrated model of vascular calcification [47]. Their investigations highlighted that the co-culture of apoE-/-EC and apoE-/-VSMC increased intracellular calcium content and ALP activity, promoted OPN and BMP-2, and inhibited OPG and MGP protein expression in ECs and VSMCs, suggesting that the co-culture of apoE-/-EC and apoE-/-VSMC had the highest risk of vascular calcification, and apoE-/-EC and apoE-/-VSMC could also accelerate calcification of normal WT-VSMC and WT-EC, respectively, through chemical mediators between ECs and VSMCs. An alternative system for studying the ECs-VSMCs connection uses a conditioned cell medium to treat a recipient cell culture. Bouabdallah and colleagues exposed human umbilical vein ECs to Pi and indoxyl sulfate and then used the conditioned media to treat recipient AoSMCs [48]. They showed a massive calcification in AoSMCs due to procalcifying factors such as IL-8, secreted from ECs in response to calcifying conditions.

### 3.3. MSCs and VSMCs Co-Cultures

The connection between bone tissue and the cardiovascular system has been demonstrated and is of great interest among researchers. Different in vivo studies showed the ability of MSCs to migrate from the circulation into vessels and differentiate into osteogenic cells, thus promoting VC [7,49]. Meanwhile, Zhu and colleagues reported the protective role of MSCs in VSMC osteogenic differentiation using a transwell co-culture system [50]. They obtained calcified VSMCs treating cells with an osteogenic medium containing 0.1 µM dexamethasone, 10 mM sodium β-glycerophosphate, and 0.05 mM ascorbic acid-2-phosphate. After 14 days of culture, a significant increase in OPG and OPN mRNA expression and increased levels of Wnt5a, Ror2, and β-catenin in VSMCs under calcifying conditions were observed. Indirect contact with MSCs could reduce calcification by paracrine and immunomodulatory effects, including the reduction of OPG and OPN mRNA levels and the downregulation of Wnt ligands (Wnt5a, Ror2, and β-catenin) in calcified VSMCs. This data divergence could be due to dual actions of MSCs that, in their de-differentiated state, might produce signals able to prevent VC in a paracrine manner while differentiating in osteoblastic cells could activate other pathways and release calcifying signals. Due to the discrepancy between in vivo and in vitro data, further investigations are needed to clarify the exact role of MSCs in VSMC osteogenic differentiation and, globally, in VC.

### 3.4. Extracellular Vesicles and Particles and VSMC Cultures

Extracellular vesicles (EV), matrix vesicles (MV), and extracellular particles (EVP) have emerged as active messengers in intercellular communication, carrying important molecular biomarkers and mediators, and assuming a relevant role for target cell and organ physiology in both health and disease as, in our case, for vascular calcification.

In the synthetic phenotype, VSMCs are characterized by the secretion of multiple soluble mediators (i.e., enzymes and pro-inflammatory cytokines) and EVs that act as nucleation nodes for VC [51].

Pan and colleagues founded that exosomes, the smallest subtype EVs, released from calcified mouse VSMCs could induce marked calcification in recipient mouse VSMCs [52]. The calcifying potential of exosomes was also highlighted by Li and colleagues in a high-glucose-stimulated human umbilical vein endothelial cells (HUVECs) model [53]. Kapustin and colleagues demonstrated that VSMC-derived exosomes calcify in vitro in response to high calcium levels, which induced the loss of calcification inhibitors and the surface exposure of phosphatidylserine, promoting their mineralization properties by forming nucleation complexes [54].

The production of VSMC-derived exosomes during calcifying conditions in vitro is regulated by sphingomyelin phosphodiesterase 3 (SMPD3) upregulation, and SMPD3 in-hibition reduces exosome release and calcification [54,55,56].

Moreover, EVs isolated from patients with end-stage CKD were used to evaluate their modulatory effects on ECs and VSMCs, and to their calcification potential. Their results highlighted angiogenesis inhibition, increased endothelial apoptosis, and enhanced VSMC calcification. Recently, Alique and colleagues demonstrated that extracellular vesicles obtained from HUVECs treated with uremic toxin and indoxyl sulfate induced calcification in recipient VMSCs [57,58].

Matrix vesicles are cellular-released vesicles isolated from the extracellular matrix containing proteins, carbohydrates, lipids, DNA, and mRNAs (Figure 2). During VC, VSMCs release calcifying matrix vesicles with a proteomic profile similar to osteoblastic vesicles acting as nucleating foci to initiate microcalcification [59]. Additionally, MMP-2, MMP-3, MMP-9, and MMP-13 are located in matrix vesicles membranes and play a key role in matrix remodeling and propagation of mineralization [60,61]. Considering the role of vesicles in cell-cell communication, co-cultures of VSMCs and matrix vesicles have been explored to understand their involvement in the calcification process. Chen and colleagues demonstrated that matrix vesicles isolated from VSMCs in CKD rats could promote the calcification of recipient VSMCs from normal rats [62]. Interestingly, both matrix and media vesicles were endocytosed by recipient normal VSMCs, but only the first increased Ca^2+^ content.

Among EVP and their involvement in VC, a relevant role is assumed by calciprotein particles (CPPs), which are colloidal circulating nanoparticles formed of calcium phosphate and serum protein fetuin-A [63,64,65,66]. Albeit CPPs receive increasing attention among researchers and clinicians, the mechanism of how these particles contribute to vascular calcification remains elusive. In the last few years, some studies used in vitro systems based on VSMC cultures to investigate the calcifying potential of CPPs and the possible molecular targets involved [67,68,69]. Braake and colleagues synthesized and isolated CPPs to evaluate their pro-calcifying properties using an in vitro model based on human AoVSMC [69]. For that purpose, cultured cells were treated with CPPs, and the quantification of total intracellular Ca^2+^ deposition was performed. The intracellular calcification induced by CPP2 is already present 24 h after induction, while those related to high Pi only occurred after 1 week. These data support the pivotal role of CPP in inducing VSMC calcification rather than soluble Pi alone. Similar results were obtained by Aghagolzadeh and colleagues [67]. In this study, CPPs were generated using a Pi-enriched culture medium and used to treat human VSMCs. Exposure of VSMCs to CPPs conditioned medium led to prominent intracellular calcification and oxidative stress induction. Moreover, they demonstrated that CPPs induced VSMC calcification through TNF-α upregulation and the stimulation of its receptor type 1. To investigate the involvement of serum CPPs and EVs in VC, Viegas and colleagues treated VSMCs with a high concentration of Ca^2+^ and Pi in the presence of human serum from CKD patients and evaluated the effect on VSMC differentiation, calcification, and inflammation [64]. Whole serum from CKD patients increased expression of the osteogenic markers Runx2 and OPN; conversely, serum depleted of these nanostructures significantly reduced VSMC calcification. Serum deprived of CPPs and EVs induced an increase in smooth muscle alpha-actin (αSMA) levels and a reduction of the osteogenic markers Runx2 and OPN, together with decreased levels of the inflammation markers interleukin-1β (IL-1β) and cyclooxygenase-2 (COX2).

These data indicate that VSMC calcification is an active process cell-mediated, related to the interaction and processing of extracellular vesicles and particles by VSMCs. Further investigation is needed to elucidate the molecular mechanisms involved.

## 4. Anticalcifying Agents

We searched the existing literature for compounds with counteracting effects on calcification in in vitro systems and the potential mechanisms involved. The selected anticalcifying agents are classified into natural and synthetic compounds and presented in Table 2.

### 4.1. Natural Compounds

Vitamin E is a tocopherol exhibiting strong antioxidant properties on cell membranes, thus preventing the propagation of free radical reactions [70]. Huaizhou and colleagues demonstrated that vitamin E attenuated intracellular oxidative stress and reduced the calcium content and osteoblast differentiation of human AoSMCs by inhibiting the ERK pathway [71].

Vitamin K is a fat-soluble vitamin that occurs in two forms: vitamin K1 (phylloquinone) and vitamin K2 (menaquinones) [91]. Several studies have demonstrated that vitamin K is involved in VC through the carboxylation of matrix Gla protein (MGP), which acts as a potent calcification inhibitor [92,93,94,95]. Cuiting and colleagues demonstrated the protective role of vitamin K2 (10 μM) against CaCl_2_- and β-glycerophosphate-induced VSMC calcification by preventing apoptosis via restoration of the Gas6-Axl signaling pathway [72].

The natural flavonol quercetin is enriched in many fruits and vegetables and is widely used as a dietary supplement [73]. Quercetin completely nullified warfarin-induced aortic calcification by inhibiting transglutaminase 2 and β-catenin expression in vitro [74], restored mitochondrial integrity with improved membrane potential and ATP production, reduced fragmentation/fission, and reversed the apoptotic process in calcified VSMCs [75]. Recently it was published that quercetin attenuated oxidized low-density lipoprotein-induced VSMCs calcification, reduced ALP activity, down-regulated the expression levels of BMP2 and Osterix, and upregulated the expressions of vascular smooth muscle contractile proteins α-SMA and SM22a–partially restoring the contractile phenotype. Quercetin also increased SOD activity and reduced ROS levels by targeting the TLR4 signaling pathway [76].

Carnosine is a dipeptide with antioxidant, anti-inflammatory, and anti-senescence properties [77,78]. Yi and colleagues showed that carnosine inhibited Pi-induced VSMCs calcification by reducing Ca^2+^ levels and ALP activity and suppressing the expression of the osteogenic markers Runx2 and BMP-2 [79]. Carnosine also inhibited the mTOR pathway [80], which plays a crucial role in the osteoblastic differentiation of VSMCs [96].

Magnesium (Mg^2+^) has recently been identified as a new player in the VC process. Several studies showed that Mg^2+^ supplementation alleviates phosphate-induced calcification in in vitro VSMCs [81]. These studies showed that Mg2+ influenced the calcification process through transient receptor potential melastatin activity and by modulating the expression of anti-calcification proteins (OPN, MGP), osteogenic proteins (Osteocalcin, BMPs), and transcription factors (Cbfa1/Runx2 and Osx).

Intermedin (IMD), also known as adrenomedullin 2, is a secreted peptide of 148 amino acids belonging to the calcitonin gene-related peptide superfamily [82]. The proteolytic cleavage generates IMD1-53 at Arg93-Arg94, the main active fragment of IMD [83]. Jin-Rui and colleagues demonstrated that IMD1–53 reduced the expression of proteins related to ER stress (e.g., activating transcription factor 4 and 6, glucose-regulated protein 78 and 94), obtained with tunicamycin or dithiothreitol treatment, though cAMP/PKA signaling. In addition, IMD1–53 restored the loss of VSMC contractile markers and ameliorated calcium deposition and alkaline phosphatase activity in calcified VSMCs [84]. Interestingly, Yao and colleagues elucidated the effects of IMD1-53 on VSMC osteogenic differentiation related to reducing p16 and p21 expression in vivo and in vitro [84]. In addition, IMD1-53 upregulates the mRNA and protein expression of sirt1 by activating PI3K/Akt, AMPK, and cAMP/PKA signaling, indicating that IMD1-53 protected against intracellular calcification by upregulating sirt1 via PI3K/Akt-AMPK-cAMP/PKA axis.

### 4.2. Synthetic Compounds

Many synthetic compounds exhibit antioxidant properties and could effectively counter intracellular calcium deposition during VC. Acetylcysteine is the best known and acts on hepatic glutathione synthesis [86]. Firstly, acetylcysteine is oxidized to diacetylcystine, whose complete metabolism leads to several metabolic products such as cysteine, cystine, inorganic sulfate, and glutathione. Bourne and colleagues demonstrated that acetylcysteine could reduce VSMC cell death and calcification through its antioxidant and anti-inflammatory properties [87]. VSMCs display reduced GSS, GCL, and GSH expressions in the calcifying environment. Interestingly, acetylcysteine treatment increased GSH levels in VSMCs, restoring the oxidoreductive imbalance due to ROS accumulation [86].

Son and colleagues investigated the potential anticalcifying properties of statins, such as pravastatin, atorvastatin, and fluvastatin [88]. Researchers found a significant reduction in human AoSMCs Pi-induced calcification following statins treatment (reduced by 49% at 0.1 μmol/L atorvastatin). Demin and colleagues demonstrated that atorvastatin suppressed calcium accumulation and osteogenic differentiation in TGF-β1-stimulated VSMCs by inducing autophagy through downregulation of β-catenin signaling, suggesting the potential role of autophagy in vascular pathophysiology [89]. The inhibitory effect of statins was related to apoptosis prevention, restoring the survival pathway mediated by Gas6–Axl axis that is down-regulated during VC [19].

Dai and colleagues found that the autophagy inhibitor 3-methyladenine significantly promoted the release of calcified vesicles, increased ALP activity, and calcium deposition in bovine AoSMCs subjected to high Pi levels [22]. In contrast, the autophagy inducer valproic acid significantly attenuated phosphate-induced calcification in VSMCs, confirming the role of autophagy in inhibiting Ca^2+^ deposition [90].

## 5. Conclusions

Multiple factors regulate VC, and, in this pathological process, VSMCs play a central role. We reported in Figure 3 the network analysis obtained with the integration of data published so far. This analysis highlighted 49 interconnected genes, and among these, several are directly related to the regulation of smooth muscle cell proliferation, response to oxidative stress, regulation of inflammatory response, and autophagy. Further investigations are required to understand the detailed molecular mechanisms to counteract VC and its related medical complications pharmacologically. The variety of factors that influence the development and progression of VC, and the inadequate transferability of animal data to the human setting, justifies the choice of human in vitro models as a valuable platform for pathophysiologic investigations. Nevertheless, the complexity of VC, in terms of the many different cell types involved and the multitude of cellular processes, makes choosing the in vitro models challenging. One single in vitro model could not replicate the VC, and several models may be needed to account for a wide spectrum of issues, a multifaceted system characterized by different cell types and cellular processes. Due to the prominent role of VSMCs in VC initiation and development, many studies have focused on these cell types to develop disease-relevant models. Starting from this assumption, in vitro assays may be used to examine their connection to endothelial damage and the potential impact on VC initiation and progression at the molecular level. Co-culture systems utilizing VSMCs and other cell types (i.e., endothelial and mesenchymal cells) have been developed using several approaches. These include monoculture, direct co-culture or growing cells on the opposite sides of the membrane (transwell systems). In vitro models can also be used as high-throughput screening platforms for novel pharmacological agents potentially able to inhibit VC. Some natural and synthetic compounds that showed promising anticalcifying properties in vitro are reported in Table 2. The nature of an in vitro system can permit multi-endpoint analyses in a single experiment. These include, for example, the possibility of measuring the expression of secreted proteins or quantifying metabolites in the culture media, localizing proteins using immunohistochemistry, and performing functional assays. Notably, the expression of several genes and proteins is altered in endothelial and muscle cells exposed to the calcifying environment and is directly related to calcification processes in humans. Thus, these genes could also be used as biomarkers, enhancing the relevance of data from in vitro models. Regardless of the actual model being used, one potential criticism of in vitro systems, especially for those regarding cardiovascular pathologies, is the nature of their culture under static conditions. In humans, vascular cells are exposed to mechanical stimuli imparted on the vessel wall by the flowing blood. Advanced in vitro models would allow for the exposure of cells under flow conditions that better mimic those in vivo.

## Figures and Tables

**Figure 1 biomedicines-10-02491-f001:**
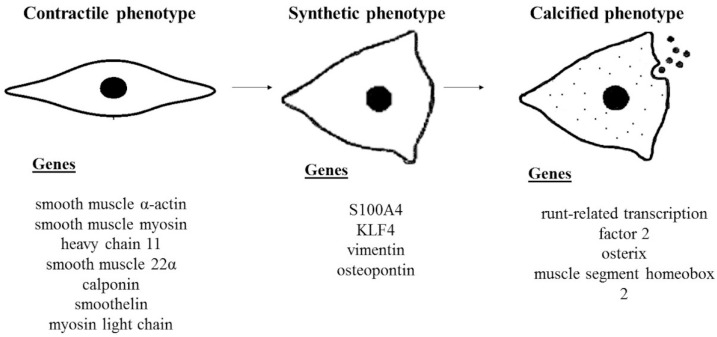
VSMCs switch from contractile to calcified phenotype. VSMCs exhibit a differentiated contractile phenotype characterized by a repertoire of smooth muscle-specific genes/proteins and an elongated, spindle-shaped morphology. In response to different signals and stimuli, VSMCs switch to a dedifferentiated synthetic phenotype with rhomboid morphology. In this status, VSMCs are characterized by increased proliferation rate, migration, and upregulation of genes mentioned above. Synthetic VSMCs in a calcifying environment may switch to an osteoblastic-like phenotype characterized by acquiring osteogenic properties, increasing arterial rigidity.

**Figure 2 biomedicines-10-02491-f002:**
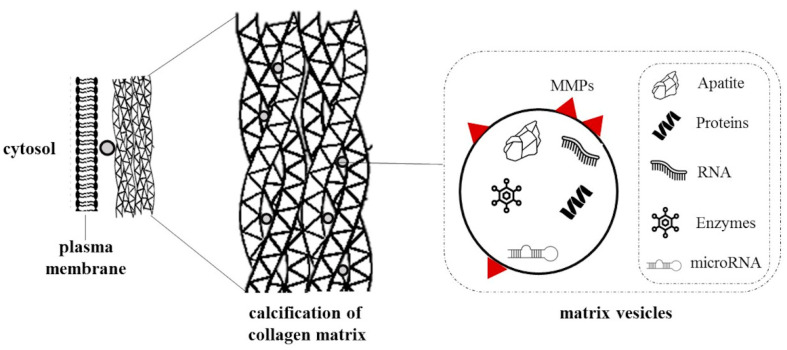
Schematic representation of matrix vesicles. Matrix vesicles are peculiar extracellular matrix vesicles that contain various cargoes, including proteins, carbohydrates, lipids, enzymes, RNA, and apatite. Evidence showed that extracellular matrices serve as nucleating foci to initiate microcalcification.

**Figure 3 biomedicines-10-02491-f003:**
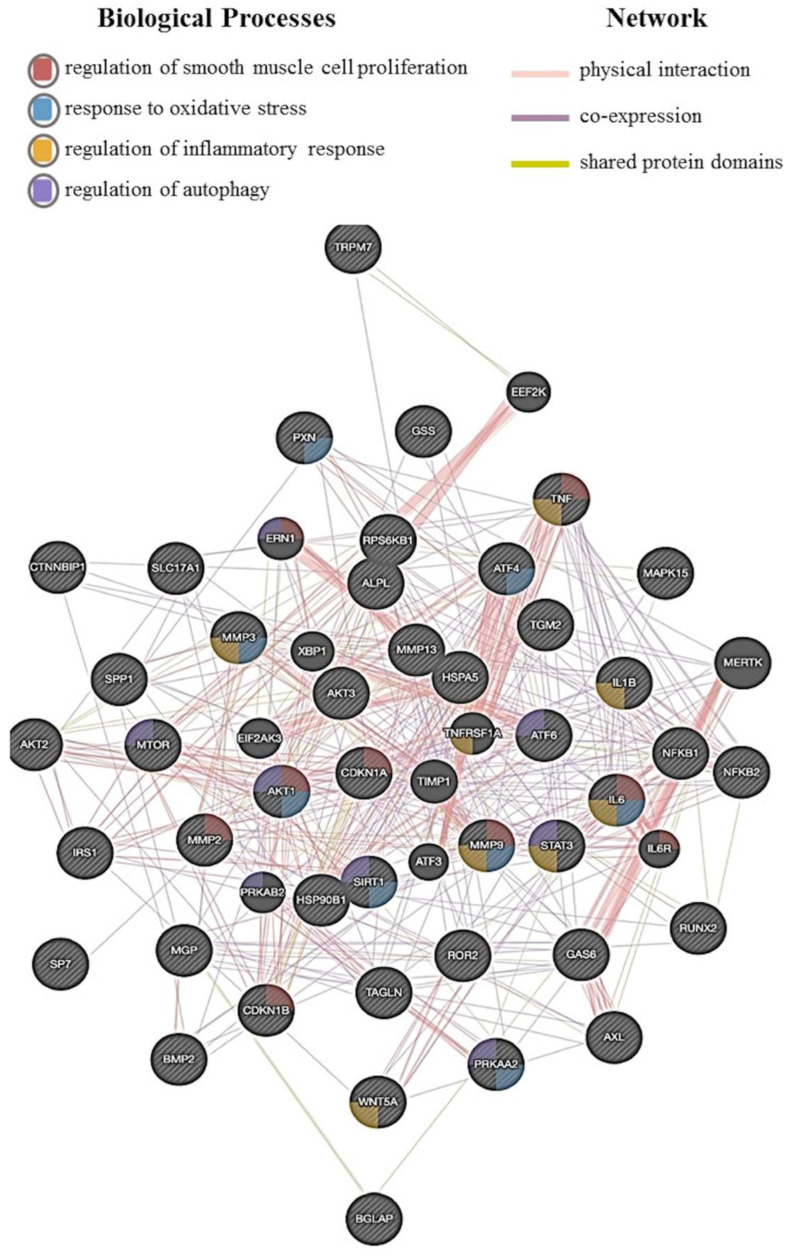
A network analysis of gene involved in VSMCs calcification. The network shows the correlations between genes. Each node represents a gene, and the relationship between two different genes is described with a line (physical interaction, co-expression, and shared protein domains). The analysis highlighted 4 biological processes relevant to VSMC calcification: regulation of smooth muscle cell proliferation, response to oxidative stress, regulation of inflammatory response, and autophagy.

**Table 1 biomedicines-10-02491-t001:** Experimental conditions for inducing calcification in in vitro systems.

Cell Type	Calcifying Medium	References
Bovine VSMC	DMEM high glucose, 15% FBS, 10 mmol/L sodium pyruvate, 10 mmol/L β-glycerophosphate, 10^−7^ mol/L insulin, 50 μg/mL of ascorbic acid	[32]
Mouse VSMC	DMEM, 15% FBS, 10 mmol/l sodium pyruvate, 50 mg/mL ascorbic acid, 10 mmol/l β-glycerophosphate	[33]
Mouse VSMC	DMEM high glucose, 15% FBS, 10 mmol/L β-glycerophosphate, 284 µmol/L ascorbic acid, 10 mmol/L sodium pyruvate	[34]
Bovine VSMC	DMEM high glucose, 10% FBS, 10 mM β-glycerophosphate, 50 μg/mL ascorbic acid, 25 mM glucose	[35]
Mouse VSMC	DMEM, 1% FBS, 1.6 mM inorganic phosphate, 10 μM warfarin	[36]
Mouse VSMC	DMEM high glucose, 15% FBS, 8 mmol/l CaCl_2_, 10 mmol/l sodium pyruvate, 1 μmol/l insulin, 50 μg/mL ascorbic acid, 10 mmol/l β-glycerophosphate, and 100 nmol/l dexamethasone	[37]
Mouse VSMC	DMEM, 10% FBS, 2.5 mmol/L CaCl_2_, 5 mmol/L β-glycerophosphate	[38]
Mouse VSMC	DMEM high glucose, 10% FBS, 1.25/2.5 mM β-glycerophosphate, 25/50 µg/mL ascorbic acid	[39]

**Table 2 biomedicines-10-02491-t002:** Relevant natural and synthetic compounds with anticalcifying properties in in vitro models.

Natural Compounds	References
Vitamin E	[70,71]
Vitamin K	[72]
Quercetin	[73,74,75,76]
Carnosine	[77,78,79,80]
Magnesium	[81]
Intermedin	[82,83,84,85]
**Synthetic Compounds**	
Acetylcysteine	[86,87]
Statins	[37,88,89]
Valproic acid	[90]

## Data Availability

Not applicable.

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
