# Peer review of "Vascular Calcification: In Vitro Models under the Magnifying Glass"

_biomedicines, 2022, doi:10.3390/biomedicines10102491_

Round 1
Reviewer 1 Report
Dear Authors,
Thank you for presenting a very good manuscript; Overall, it was a pleasure to read this manuscript!
I have only minor comments, included within the file attached.
As a general comment, the manuscript is well prepared and easy to read; The figures are clear, eventhough the letter size in the legends being quite small; I would suggest bigger letters in the legends
Congratulations on your work! I guess it will be quite useful for researchers or clinitians or anyone looking for a better understanding of the physiopathology of vascular calcification as well as the relevance of "in vitro" models of the disease.

Author Response
Dear reviewer,
Thank you for the valuable suggestions. We have revised the manuscript following your comments, and have greatly improved the quality of our manuscript. In addition, we have carefully revised the English language and corrected some grammatical errors. The answers and comments to your requests are below.
- Line 56: According to your suggestion, we changed the sentence to make it clearer.
- Line 169: As you suggested, we specified the acronym MMP.
- Line 201-206: The role of mesenchymal stem cells appears very complex. According to your suggestion, we have tried to clarify the concepts as follows:
“This data divergence could be due to dual actions of MSCs that in their de-differentiated state might produce signals able to prevent VC in a paracrine manner while differentiating in osteoblastic cells could activate other pathways and release calcifying signals. Due to the discrepancy between in vivo and in vitro data, further investigations are needed to clarify the exact role of MSCs in VSMC osteogenic differentiation and, globally, in VC”.
- The figures have been modified according to your suggestions.
We hope you consider our work adequate, and the manuscript is publishable in this form.
Reviewer 2 Report
This manuscript review aims to give the reader a general approach on the in vitro cell models reported in studies involving vascular calcification and the study of the processes and molecular mechanisms involved. The manuscript is well written but still some improvements should be made.
Importantly in this review aiming to include relevant players and agents with a role in the vascular calcification (VC) process the authors lack including the well-known association between circulating calcified nanostructures also called calciprotein particles CPP or role of circulating extracellular vesicles in VC using in vitro VSMC (Arterioscler Thromb Vasc Biol. 2018 Mar;38(3):575-587; Aging, 9(3)(2017), pp.778-789). Based on this, the section Matrix vesicles and VSMCs cultures is not complete and needs further improvement with the former current knowledge.
Also, when describing anticalcifying agents specially natural compounds, the authors should not miss including Vitamin K and its role as cofactor of gamma carboxylation and functionalization of several Vitamin k-dependent proteins (e.g Matrix Gla Protein and Gla rich Protein) with a significant role on the prevention of VC.
Table 1 needs to be reformulated. The table should list instead the complete Calcifying culture media conditions and detail which specific cell type or system was referred in the study. In its present form the information in the table can became incorrect and misleading.
Figure 1 needs improvements.
Figure 2 needs improvements and lacks relevant information
Figure 3 is not readable in its current form
Author Response
Dear reviewer,
Thank you for the valuable suggestions. We have revised the manuscript following your comments, and have greatly improved the quality of our manuscript. In addition, we have carefully revised the English language and corrected some grammatical errors. The answers and comments to your requests are below.
- Importantly in this review aiming to include relevant players and agents with a role in the vascular calcification (VC) process the authors lack including the well-known association between circulating calcified nanostructures also called calciprotein particles CPP or role of circulating extracellular vesicles in VC using in vitro VSMC (Arterioscler Thromb Vasc Biol. 2018 Mar;38(3):575-587; Aging, 9(3)(2017), pp.778-789). Based on this, the section Matrix vesicles and VSMCs cultures is not complete and needs further improvement with the former current knowledge.
R: We agree with the reviewer on the need to include extracellular vesicles and CPPs in our manuscript, as they play an important role in VC. We revised the previous paragraph, “Matrix vesicles and VSMCs cultures”, to “Extracellular vesicles and particles and VSMCs cultures”, providing a more detailed overview (lines 210-264).
- Also, when describing anticalcifying agentsspecially natural compounds, the authors should not miss including Vitamin K and its role as cofactor of gamma carboxylation and functionalization of several Vitamin k-dependent proteins (e.g Matrix Gla Protein and Gla rich Protein) with a significant role on the prevention of VC.
R: According to your suggestions, we added and discussed the role of vitamin K in the paragraph “anticalcifying agents” in the section natural compounds (lines 278-284).
- Table 1 needs to be reformulated. The table should list instead the complete Calcifying culture media conditions and detail which specific cell type or system was referred in the study. In its present form the information in the table can became incorrect and misleading.
R: We have reformulated Table 1 according to your suggestions to clarify the composition of calcifying media and the cell type used.
- The figures have been modified and added the information that you have suggested.
We hope you consider our work adequate, and the manuscript is publishable in this form.
Round 2
Reviewer 2 Report
Dear authors
The revised version of the manuscript already tries to reflect some important information in this field that was previously missing.
Nevertheless, regarding the role of EVs and calcifying protein particles in VSNCs calcification the research included is still quite incomplete and missing important information recently published in this area of research that highlight the prominent role of EVs secreted by VSMS in the dynamics/regulation of VC. namely. Roles and Regulation of Extracellular Vesicles in Cardiovascular Mineral Metabolism. Front. Cardiovasc. Med. 2018, 5, 187; Initiation and Propagation of Vascular Calcification Is Regulated by a Concert of Platelet- and Smooth Muscle Cell-Derived Extracellular Vesicles. Front. Cardiovasc. Med. 2018, 5, 36; Chronic Kidney Disease Circulating Calciprotein Particles and Extracellular Vesicles Promote Vascular Calcification: A Role for GRP (Gla-Rich Protein). Arterioscler Thromb Vasc Biol. 2018;38(3):575.
The information should be included to better address the interest of the readers in this field.
Author Response
Dear reviewer,
Thank you for the valuable comments that have greatly improved the quality of our manuscript. We have studied with particular attention the review article that you have mentioned (“Roles and Regulation of Extracellular Vesicles in Cardiovascular Mineral Metabolism”), where we found many interesting articles that allowed us to add new information regarding the prominent role of EVs in the VC process.
Our reviews are listed below:
- Section “Extracellular vesicles and particles and VSMC cultures”: we added two paragraphs from line 217 to line 227, and from line 271 to line 279.
- Section “References”: we added the new following 6 references: 44-48, 62.
We hope you consider our work adequate, and that the manuscript is publishable in this form.